# Organismal Roles of Hsp90

**DOI:** 10.3390/biom13020251

**Published:** 2023-01-29

**Authors:** Patricija van Oosten-Hawle

**Affiliations:** Department of Biological Sciences, The University of North Carolina at Charlotte, Charlotte, NC 28223, USA; pvanoost@uncc.edu

**Keywords:** Hsp90, organismal, cell nonautonomous, proteostasis, stress response, inter-tissue stress signaling

## Abstract

Heat shock protein 90 (Hsp90) is a highly conserved molecular chaperone that assists in the maturation of many client proteins involved in cellular signal transduction. As a regulator of cellular signaling processes, it is vital for the maintenance of cellular proteostasis and adaptation to environmental stresses. Emerging research shows that Hsp90 function in an organism goes well beyond intracellular proteostasis. In metazoans, Hsp90, as an environmentally responsive chaperone, is involved in inter-tissue stress signaling responses that coordinate and safeguard cell nonautonomous proteostasis and organismal health. In this way, Hsp90 has the capacity to influence evolution and aging, and effect behavioral responses to facilitate tissue-defense systems that ensure organismal survival. In this review, I summarize the literature on the organismal roles of Hsp90 uncovered in multicellular organisms, from plants to invertebrates and mammals.

## 1. Introduction

Cellular protein homeostasis depends on the integrity and function of its proteome, of which molecular chaperones play an indispensable role to maintain it. Heat shock protein 90 (Hsp90) is an essential and evolutionary conserved molecular chaperone, that except for archea, is found in all kingdoms of life [1]. Hsp90 is crucial for the viability and growth of eukaryotic cells and organisms, and it is one of the most abundant cellular proteins known to date, representing ~2% of total protein in a cell [2]. This abundance of Hsp90 is required to sustain the wide range of cellular processes it is implicated in, by chaperoning components involved in cellular signal transduction events such as protein kinases and steroid hormone receptors [3,4,5,6,7,8]. 

Hsp90 requires ATP function for its activity to help facilitate folding of client proteins, and this function is regulated and controlled by a variety of co-chaperones in a context-specific manner [7,9], as well as post translational modifications [10,11,12,13]. Briefly, each Hsp90 dimer binds ATP in its “open conformation”, which is followed by dimerization of the N-terminal domains of each protomer, allowing ATP hydrolysis. This subsequently leads to release of the folded and mature client protein, resulting in the open conformation of the Hsp90 dimer [7,8,14]. Each step along the Hsp90 chaperone cycle is finetuned by co-chaperones, such as, e.g., Cdc37/p50, which promote binding to kinase clients [15,16,17], or Aha1 and p23 that are involved in the regulation of Hsp90 ATP hydrolysis [14,18,19]. For more detailed information on the Hsp90 structure and regulation by co-chaperones, the reader is referred to articles and reviews specifically discussing this topic, including this Special Issue [8,20,21,22].

Because of its involvement in a wide range of cellular processes, Hsp90 supports an expansive network of more than 300 confirmed client proteins encompassing at least 5% of total proteins in yeast cells, and this number is similarly increased in multicellular organisms and mammals (https://www.picard.ch/downloads/Hsp90interactors.pdf; accessed 20 December 2022). Consequently, it is perhaps not surprising that Hsp90 is involved in almost every cellular process from cell cycle and a multitude of signal transduction pathways to protein trafficking, transcriptional processes and genomic stability [23,24,25,26,27,28,29]. In multicellular organisms, Hsp90’s involvement is expanded accordingly, and evidence in the past two decades has shown that Hsp90’s role reaches far beyond maintenance of signaling proteins: Its involvement ranges from development and evolution to intercellular stress signaling, aging responses and innate immunity, as well as neuronal function and behavior (Figure 1). In this review, I will highlight these organismal roles of Hsp90 which have been uncovered in different multicellular model systems, including plants, invertebrates and mammals. 

## 2. Hsp90 in Organismal Development and Evolution

### 2.1. Development

The coordination of cell proliferation and differentiation is crucial for proper development. Hsp90’s central role in growth and development is profound, as its client proteins regulate almost all phases of the cell cycle. These include PI-3/AKT, NFkB and MAP kinase pathways, which drive progression through G1/S and G2/M checkpoints through transcriptional routes converging on Cyclin D, and Cyclin B and E [30]. Hsp90 also regulates various key cell cycle regulators directly, including Cdk1, Cdk2, Cdk4 and Cdk6 [31,32,33,34]. Furthermore, check point kinases Wee-1 and Myt-1 depend on Hsp90 function [35,36,37]. Later stages of mitosis and cytokinesis also depend on Hsp90 via mitotic regulators Survivin and Aurora B [38,39]. Because of Hsp90 influencing the cell cycle at multiple levels, either directly or indirectly, Hsp90 function is indispensable not only for organismal development but also for tumor cell progression, which was recognized early on through targeted inhibition of Hsp90 function using ansamycin inhibitors such as geldanamycin and 17-AAG [2,40,41,42]. Due to Hsp90 being involved in signaling pathways promoting cancer cell progression, it has become an attractive and well-established therapeutic cancer target, with Hsp90 inhibitors being continually developed and reviewed in clinical trials in an ongoing basis [43]. 

Most eukaryotic systems have two different cytosolic Hsp90 isoforms, with the exception of *C. elegans* that has only one cytosolic isoform (HSP-90/DAF-21). In mammals and yeast, the stress-inducible Hsp90α is encoded by the gene HSP90AA1 in humans (HSP82 in yeast), and the constitutively expressed Hsp90β is encoded by HSP90AB1 in humans and HSC82 in yeast. Although both isoforms share extensive sequence identity, their cellular functions are not completely identical and this is also demonstrated in their developmental requirements in an organism. For example, the Hsp90β knockout mouse shows early embryonic lethality [44], whereas this is not the case for mice lacking Hsp90α, which are viable but exhibit a failure of spermatogenesis and become sterile [45]. Moreover, while both isoforms are mutually expressed in most tissues in the mouse, the heart and muscle were found to harbor reduced levels of Hsp90α compared to Hsp90β [45]. Interestingly, zebrafish contains two Hsp90α genes, called Hsp90a1 and Hsp90a2. Hsp90a1 is crucial for myofibril organization in skeletal muscle development, whereas Hsp90a2 has no effect on muscle development [46]. Coherent with observations in vertebrates, the only cytosolic Hsp90 isoform in the invertebrate *C. elegans* (HSP-90) is crucial for myosin folding and muscle development, as RNAi-mediated knockdown leads to disrupted myosin filaments and motility defects [47,48]. Mammals such as mice, however, appear to require higher threshold levels of Hsp90 to promote stress adaptation and survival of the organism compared to yeast. This is accomplished through an internal ribosome entry site (IRES) in the 5’UTR of the Hsp90ab1 mRNA that can reprogram Hsp90 translational levels in stressed conditions [49]. 

As observed in other multicellular model organisms, depletion of Hsp90 by RNAi in *C. elegans* leads to morphological and transcriptional changes, including developmental changes to the gonad, vulval structures and oocyte development [50]. Indeed, Hsp90 regulates the meiotic prophase to metaphase transition during oocyte development by ensuring *wee-1* kinase functionality, which results in reduced fertility in the worm [51]. Interestingly, besides transcriptional changes that demonstrate induction of the heat shock response, Hsp90 RNAi at the whole animal level also leads to induction of an innate immune response, by altering expression levels of innate immune genes primarily expressed in the intestine [50]. 

The role of Hsp90 in *C. elegans* development is further highlighted through its involvement in *dauer* formation, TGFβ- and Notch signaling. The *C. elegans* TGF-β pathway regulates a decision between reproductive development and arrest at a larval stage known as *dauer* that is suited for survival under conditions of environmental stress. Hsp90 itself is implicated in *dauer* formation through its interaction with two components of the TGF-β pathway, TGF-β-RI (DAF-1 in *C. elegans*) and TGF-β-RII (DAF-4 in *C. elegans*) [52]. Hsp90 also regulates the functionality of a DAF-11/guanyl cyclase signaling pathway in sensory ciliae and amphid neurons that controls *dauer* formation in response to environmental cues, in parallel to the TGF-β pathway [53,54], as well as chemosensory behaviors [54,55]. Germline proliferation in *C. elegans* requires signaling from the somatic gonad to the germline, which is mediated by GLP-1 (a Notch orthologue) [56]. If GLP-1/Notch signaling is defective through mutations in the *glp-1* gene, germline stem cells prematurely exit mitosis and enter meiosis to form gametes, resulting in reduced germline proliferation and sterility [57]. Hsp90 has been identified as a regulator of Notch signaling that suppresses defective GLP-1/Notch signaling and promotes germline proliferation [58]. Strikingly, *C. elegans* depleted for Hsp90 by RNA interference or using an HSP-90(I461N) mutant leads to the formation of a proximal germline tumor, despite its reduced function and reduced GLP-1 signaling [57,58]. Further research in solving this paradox will be required to better illuminate the complex tissue-specific and organismal functions of the Hsp90 chaperone system in *C. elegans* and other metazoans. 

### 2.2. Hsp90 as a Capacitor of Organismal Evolution

The physiological requirement of Hsp90 for the growth and development of model organisms was obvious early on using *Saccharomyces cerevisiae* [35,59], but was further highlighted using *Drosophila melanogaster* [60], where point mutations of the *Drosophila* Hsp90 (Hsp83) gene are lethal as homozygotes [61]. Although heterozygous mutant combinations are viable as adults, they are associated with sterility due to defects in microtubule dynamics during spermatogenesis [61]. Further experiments in *Drosophila* designed to identify suppressors of signal transduction *Sevenless* and *Raf* pathway mutants recovered Hsp90 mutants. Progeny of these Hsp90 mutants resulted in developmental abnormalities that, dependent on the genetic background, affected different morphological structures of the fruit fly [60,62]. This discovery led to further demonstration of Hsp90’s importance in *Drosophila* spermatogenesis and germline development [61]. It was one of the cornerstones that defined Hsp90’s prominent role in evolution, which was established as the “Hsp90 capacitor hypothesis” by the Lindquist lab [63]. 

The capacitor hypothesis demonstrated that diverse pathways become sensitive to the effects of genetic variation when Hsp90 function is compromised due to environmental stress, pharmacological inhibition or genetic mutation. It showed that Hsp90 functions in a wide variety of morphogenetic processes that are apparent in all model organisms tested, from yeast to vertebrates [63,64,65]. For example, the diverse phenotypes associated with Hsp90 impairment in *Drosophila* are deformed eye and thickened wing phenotypes [63], whereas in *Arabidopsis thaliana*, this leads to altered leaf and cotyledon shapes [64]. Similar consequences were observed in zebrafish upon reduced Hsp90 expression [66]. In *C. elegans*, the expression level of Hsp90 in particular varies during *C. elegans* embryonic development, causing embryos with stronger induction of Hsp90 to be less affected by mutation, thus buffering genetic variation [67]. However, individuals of a population with increased stress resistance due to higher Hsp90 levels show a “trade-off” with lowered reproductive potential, whereas worms with lower stress resistance are associated with higher reproductive fitness. The reason for this is thought to be a bet-hedging strategy, which is beneficial in ever-changing environments, that ensures survival of the population as a whole [67,68]. The reduced reproductive fitness due to increased Hsp90 expression perhaps highlights the requirement for Hsp90 expression levels to be tightly regulated due to its important role in germline development [57]. 

Importantly, phenotypic traits revealed upon temporary Hsp90 impairment can be selected for over several generations and become fixed in following generations, establishing Hsp90’s crucial role in evolution. An example for this in a natural setting was provided by the cavefish *Asyanax mexicanus*, where cryptic variation in eye size was masked by Hsp90 in the ancestral river but revealed when fish were kept in caves that challenged the Hsp90 system due to low-salinity water [65]. This even plays a role in human disease, as is the case in Fanconi anemia (FA), a complex autosomal recessive human cancer predisposition syndrome that results in point mutations of 19 genes involved in the FA genome maintenance pathway [69]. The function of less severe FANCA mutants was preserved by Hsp90 binding, which maintained FA pathway function but became destabilized and sensitive to genotoxic stress upon Hsp90 impairment [69].

However, while the evolutionary capacitor hypothesis relies on the potential of cytosolic protein instability that can be exposed upon Hsp90 inhibition, other contributions were shown to underlie Hsp90-dependent transcriptional mechanisms and chromatin structure [25,26,27,28,70,71]. While these are seemingly different mechanisms leading to the variety of Hsp90 buffered traits, it is perhaps a combination of multiple Hsp90-dependent genetic as well as epigenetic mechanisms working in concert. 

## 3. Hsp90-Dependent Regulation of Organismal Proteostasis, Stress and Aging

### 3.1. Hsp90 in the Regulation of Cell Nonautonomous Stress Signaling 

Hsp90, together with its co-chaperone machinery, is an integral part of the cellular network that safeguards proteostasis. As with other chaperones, Hsp90 expression is regulated by the stress transcription factor Heat Shock Factor 1 (HSF1) and is increased in response to environmental challenges that initiate the cytosolic heat shock response (HSR) [72]. This is accomplished in a negative feedback mechanism, whereby under normal conditions, HSF1 is sequestered by a multichaperone complex including Hsp90 and Hsp70 in an inactive monomeric form [73]. Proteotoxic stress conditions that increase the amount of misfolded proteins in the cell recruit the chaperones away from HSF1 towards the accumulating pool of misfolded proteins and releases HSF1 monomers. This in turn allows HSF1 monomers to form homotrimers that translocate to the nucleus, where they bind to heat shock elements (HSE) that induce molecular chaperones (heat shock proteins), as well as trafficking and proteolytic genes, in order to restore cytosolic proteostasis. Once the levels of Hsp90 and other chaperones have sufficiently increased in the cytosol to refold damaged proteins, they are recruited back to HSF1. 

However, expression of Hsp90 can also, directly or indirectly, be regulated by other transcription factors in addition to HSF-1. For example in *C. elegans*, the GATA transcription factor PQM-1 responds to local changes in Hsp90 expression levels as a mediator of transcellular chaperone signaling, but also regulates Hsp90 expression itself [74]. In addition to HSF1, Hsp90 also regulates the function of the FOXO orthologue DAF-16 isoform A by facilitating its translocation into the nucleus upon heat stress and reduced ILS [75].

In metazoans, the stress-dependent induction of HSF-1 transcriptional activity also depends on intercellular stress signaling responses. In *C. elegans*, temperature alterations are sensed by two thermosensory AFD neurons that control temperature-dependent behaviors. This is accomplished through the action of the guanylyl cyclase GCY-8 that is specifically expressed in AFD neurons, and which controls HSF1-dependent induction of the HSR in distal cells in order to restore proteostasis at the organismal level. Neuronal control of proteostasis in response to acute temperature challenges is, however, uncoupled from aging-related responses via a GPCR thermal receptor GTR-1 expressed in chemosensory neurons [76]. The *C. elegans* nervous system relays the signal to distal organs via the neurotransmitter serotonin, thus involving serotonergic neurocircuitry [77,78]. However, astrocyte-like cells in the nervous system can also regulate the cell-nonautonomous HSR in an HSF-1 dependent manner that does not rely on known neurotransmitters but instead requires small clear vesicle release [79]. Non-neuronal tissues such as muscle and gut cells can equally relay information of temperature changes to thermosensory AFD neurons via estrogen signaling through the nuclear hormone receptor NHR-69 [80], an orthologue of the human HNF4 transcription factors that are clients of Hsp90 [81].

Interestingly, heat shock leads to rapid induction of HSF1 activity in the *C. elegans* germline [77,82,83] and HSF1 is required for gametogenesis in invertebrates and vertebrates [72,84]. Like Hsp90, HSF1 is required for germline proliferation and fecundity, relying on Insulin/IGF-1 signaling in the soma that nonautonomously activates HSF-1 in the germline [85], although whether Hsp90 is involved in this regulation is currently unknown. 

However, Hsp90 is itself involved in relaying signals from one tissue to another, particularly when its expression levels are altered in the gut or the nervous system, an organismal stress signaling response known as Transcellular Chaperone Signaling (Figure 2) [86,87]. Enhancement of Hsp90 capacity in the gut or the neurons leads to a compensatory transcriptional inter-tissue response, regulated via the transcription factor PQM-1, that induces Hsp90 expression in other distal cell types and primarily muscle cells [74,88]. This protects against the age-associated debilitating consequences of misfolded proteins expressed in muscle cells, including human amyloid beta protein or endogenously expressed metastable myosin [74,88]. How this transcriptional response is relayed from one tissue to another, however, depends on tissue context. Transcellular chaperone signaling from neurons to the muscle requires glutamatergic signaling and relies on the c-type lectin *clec-41* that associates with AMPA receptor in glutamatergic neurons (Figure 2A) [74]. Increased Hsp90 expression in the gut is relayed via the secreted immune peptide *asp-12* which leads to transcriptional upregulation of Hsp90 in muscle cells (Figure 2A) [74]. 

On the other hand, when Hsp90 levels are reduced by tissue-specific RNA interference in the gut, a compensatory signaling mechanism elevates Hsp70 expression in distant cells (Figure 2B). This is, however, not mediated by a mechanism that relies on HSF1 to activate a canonical HSR, but depends on a homeodomain transcription factor, CEH-58. HSF1 transcriptional activity is suppressed upon gut-specific Hsp90 depletion, and induction of Hsp70 relies on a different intercellular signaling cue involving TXT-1, a membrane-associated guanylate cyclase that relays the signal received from the intestine to the muscle cell nucleus where the homeodomain transcription factor CEH-58 induces Hsp70 expression (Figure 2B) [87]. Thus, there is a difference in intercellular-signaling components which depend on the tissue-type perceiving altered Hsp90 expression levels. This argues for multiple and complex layers of responses that cannot be answered by one particular molecular mechanism, at least not in a multicellular organismal setting [87]. This demonstrates that in metazoans, local Hsp90 capacity can regulate organismal proteostasis and stress resilience via Transcellular Chaperone Signaling. 

Comparable organismal effects as a result of local induction of the HSF-1 mediated HSR is also observed in mammals via neuroendocrine signaling. For example, rats undergoing restraint stress have higher cortisol levels secreted by the pituitary gland which signals to activate HSF-1 in the adrenal glands in the kidney to induce Hsp70 expression [89], although how Hsp90 itself could potentially be involved in this response is currently not known and will require further research. 

One question that often arises is whether Hsp90 is secreted as part of inter-tissue stress signaling in an organism. Hsp90 secretion has been observed in tissue-culture in response to a variety of stress conditions as well as in cancer cells [90]. Clinically, skin injury promotes Hsp90α secretion and potentiates wound healing in tissue-culture, pigs and dogs [91,92]. However, secreted, extracellular Hsp90 has not been observed as a signaling component itself involved in inter-tissue stress signaling in an organism. In fact, secretion of Hsp90 was not detected in *C. elegans* overexpressing Hsp90 in different cell types [88]. For a more detailed review on the roles of secreted, extracellular Hsp90, the reader is referred to reviews by Li and colleagues [93] in this Special Issue on Hsp90. 

### 3.2. Hsp90-Dependent Regulation of Lifespan and Aging

Consistent with a growth-promoting role, substantial depletion of Hsp90 by RNAi-mediated knockdown can lead to growth defects and larval arrest, and even shorten lifespan [50,75]. The developmental defects associated with Hsp90 RNAi are morphological changes to the gonad and vulva, induction of the HSR and changes to the muscle ultrastructure [50]. Importantly, however, mild impairment of Hsp90 either by RNAi or pharmacological inhibition leads to lifespan extension and enhances health span [94]. This was shown in a pharmacological geroprotector screen using *C. elegans* that identified two Hsp90 inhibitors, Tanespimycin and Monorden, that extended lifespan and improved health of the nematode throughout the course of aging [94]. The study found that both inhibitors acted through HSF1 to induce the age-defying and health span-inducing effects in the worm. This is consistent with HSF1’s role in promoting longevity [95,96,97]. Similar to mild Hsp90 impairment by inhibitors, moderate depletion of Hsp90 RNAi in the gut also enhances lifespan and stress resilience in *C. elegans* without any developmental issues [87]. Similar observations were made in vertebrates, where transient knockdown of Hsp90 during embryonic development in zebrafish results in cold stress resistance in adult animals [98]. Interestingly, the Hs90 co-chaperone p23 acts in key longevity pathways to regulate lifespan in a temperature-dependent manner [99]. At elevated temperatures, p23 mutation extends lifespan through DAF-16 and HSF1 signaling pathways. Short-lived phenotypes depend on the DAF-12 steroid receptor signaling pathway [99], with DAF-12 being a type II nuclear receptor that resembles the human thyroid receptor and is a known client of the p23-HSP90 complex [100]. Apart from being involved in the key longevity pathways, ILS/IGF-1 signaling and HSF-1 signaling pathways, Hsp90 is also involved in the regulation of SIRT1 in both *C. elegans* and mammalian cells [101]. Thus, Hsp90 is unique, as it is a major facilitator that ensures the efficacy of all signaling processes maintaining organismal health and promoting survival. 

## 4. Pathogen Response and Innate Immunity

The involvement of Hsp90 in immune responses is manifold, as it is implicated in the adaptive as well as innate immunity pathways in almost all organisms. In plants, R proteins are client proteins of Hsp90, which is important for the defense response against microbial pathogens [102,103]. The activation of R proteins results in local cell death to limit pathogen proliferation. Because of this, R protein activation also needs to be tightly controlled to avoid tissue damage, which is regulated by Hsp90 [102,103]. 

In the invertebrate *C. elegans*, which does not have an adaptive immune response, Hsp90 plays an important role in the innate immune response via HSF1. For example, mutant Hsp90, as well as heat shock, causes release of HSF1 from Hsp90, resulting in HSF1 initiating expression of antimicrobial peptide genes [104,105]. Coherently, depletion of Hsp90 by RNA interference also induces an innate immune transcriptional response that was proposed to be similar to the immune response after *C. elegans* exposure to *Pseudomona aeruginosa* [50]. Similarly, pathogen-infected wax moths treated with Hsp90 inhibitor 17-DMAG were protected by an increased immune response [106]. This breadth of Hsp90 being implicated in a process that mediates innate immunity via HSF1 activation demonstrates the importance and conservation of Hsp90 in the innate immune response. In mammals, Hsp90 is implicated in the presentation of antigen to T-cells and activation of macrophages [107]. Hsp90 mediates antigen presentation in target antigen-presenting cells (APC) by facilitating endocytosis of bound polypeptides [108,109]. These generated antigenic peptides are presented to MHC-I/II by Hsp90 [110]. Extracellular Hsp90 can also bind to peptide antigens to facilitate uptake of the Hsp90 antigen complex by endocytosis [107]. After the antigen is internalized, intracellular Hsp90 facilitates further processing of these peptides to the proteasome for degradation [107]. Interestingly, Hsp90 also regulates the reactivation of the human immunodeficiency virus (HIV-1) via regulation of the PKC/ERK MAPK pathways, which influences replication and gene expression of the virus [111]. In the response to pathogens, extracellular Hsp90 can act as a damage-associated molecular pattern (DAMP) signal that regulates the production of cytokines in response to pathogenic infection and inflammation [112]. This involvement of p38 and ERK MAPK pathways in response to pathogens was shown to require Hsp90 for their function through direct interaction of Hsp90 with MAP kinases p38 and ERK in evolutionary diverse organisms [16,113,114]. Hsp90 also plays an important role in growth, development and virulence of parasitic pathogens itself, such as the parasitic protozoa *Plasmodium falciparum* [115] and *Toxoplasma gondii* [116,117]. This makes Hsp90 a high-value drug target to inhibit the parasite’s growth and infection cycle in humans [117]. In summary, the role of Hsp90 in the adaptive and innate immune response is vast, and the reader is referred to specialized reviews on this topic for more detailed information (e.g., [110,118]).

## 5. Neuronal Signaling and Behavior

Considering the wide range of client proteins dependent on Hsp90 function, it is perhaps unsurprising, but nevertheless fascinating, to find it involved in neuronal signaling and function. Some of the first experimental evidence demonstrating a role for Hsp90 in neuronal function stems from research in *C. elegans*. Hsp90 is crucial for the function of specific chemosensory amphid neurons required to sense pheromones and other attractants. It was proposed that Hsp90 accomplishes this through interaction and stabilization of the transmembrane guanylyl cyclase DAF-11, which regulates cGMP levels, a prominent second messenger in *C. elegans* chemosensory transduction [54,55]. 

In mice, Hsp90 is required for the constitutive trafficking of glutamatergic AMPA-type receptors into synapses during their continuous cycling between synaptic and non-synaptic sites, as well as efficient neurotransmitter release at the presynaptic terminal [119]. In addition to its role in neuronal signaling, Hsp90 chaperones the pro-regenerative dual leucine zipper kinase (DLK), a critical neuronal sensor that drives axon regeneration, degeneration and neurological disease in Drosophila and mammalian neurons [120]. This suggests a vital role for Hsp90 in axon injury signaling, as well as neuronal function that is evolutionary conserved in both vertebrates and invertebrates.

With this importance for neuronal signaling, is it possible that Hsp90 could be involved in the regulation of behavioral responses that facilitate survival during stress conditions? There is at least one example in *C. elegans* that provides direct evidence supporting such a role. Exposure of nematodes to high concentrations of volatile compounds, such as benzaldehyde and diacetyl, induces toxicity and food avoidance behavior [121]. However, preconditioning with benzaldehyde activates stress responses mediated via DAF-16, SKN-1 and HSP-90 in non-neuronal cells that confer increased stress resilience and behavioral tolerance [121]. Another example is provided by the heat stress-induced activation of HSF1, which regulates behavioral responses through estrogen signaling from non-neuronal cells to thermosensory neurons [80]. *hsf-1* mutants are defective in their thermotactic response towards temperature, i.e., migration towards cultivation temperature. Expression of wild type HSF-1 in muscle or intestinal cells rescued this behavioral defect via activation of the NHR-69 nuclear hormone receptor involved in estrogen-like signaling [80], which is a client of Hsp90, as mentioned earlier. Thus, Hsp90, through its role in multiple stress-responsive signaling pathways, may influence behavioral outputs in order to promote survival during environmental stress conditions.

## 6. Outlook and Conclusions

As a chaperone safeguarding the functionality of clients involved in almost every cellular signaling process, Hsp90 is essential for cellular homeostasis. At the organismal level, intercellular signaling processes that require the involvement of Hsp90 may be underlying the organismal coordination of extra- and intracellular signaling networks between and across different tissues and organs. Especially at the organismal level, many open questions remain to fully comprehend the organismal biology of Hsp90, particularly with regard to intercellular stress signaling. 

For example, (1) is there is a tissue map or tissue hierarchy allowing highly coordinated signaling responses to occur? We know that stress signaling can be regulated via both the nervous system and non-neuronal cell types, with, e.g., muscle and gut cells transmitting feedback information to the nervous system or even suppressing stress responses in different cell types and organs. (2) What is the tissue-specific Hsp90 interactome in an organism and how are potential Hsp90 interactors of these tissue-specific networks contributing to intercellular stress signaling? (3) Is there a role of extracellular Hsp90 in intercellular signaling processes? (4) If the tissue-specific expression levels of Hsp90 can affect stress responses in distant tissues, is there a naturally occurring/physiological condition that alters Hsp90 expression levels to induce transcellular chaperone signaling? (5) As Hsp90 function is tightly regulated by co-chaperones and post translational modifications [13], we currently do not know how co-chaperones of the Hsp90 machinery and its PTMs are involved in organismal proteostasis. For example, it can be envisioned that stress responses and intercellular stress signaling pathways are similarly influenced and perhaps finetuned through tissue-specific PTMs and co-chaperone networks. (6) How do the organismal roles of Hsp90 affect diseases, including neurodegenerative diseases and cancer, in a tissue- and disease-specific context? 

Thus, the involvement of Hsp90 in almost all aspects of organismal biology, from development to aging, stress adaptation, evolution and different diseases including cancer and neurodegenerative diseases, places it at the nexus of a plethora of cell nonautonomous signaling processes. The challenge for future research will be to navigate through these inter-tissue signaling pathways in a comprehensive manner to understand their increased complexity in the multicellular setting of an organism. 

## Figures and Tables

**Figure 1 biomolecules-13-00251-f001:**
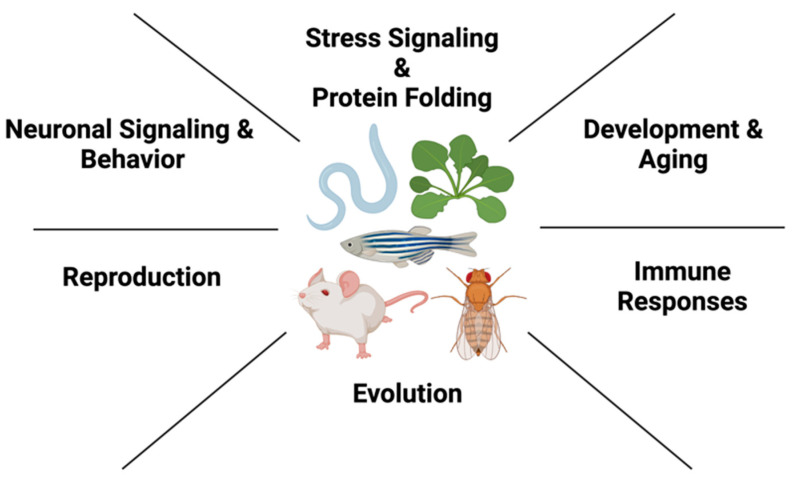
Organismal roles of Hsp90 in different multicellular model systems. In metazoans, such as *C. elegans*, *Mus musculus*, *D. melanogaster*, *Danio rerio* and *A. thaliana*, Hsp90 acts in diverse biological processes to ensure organismal proteostasis.

**Figure 2 biomolecules-13-00251-f002:**
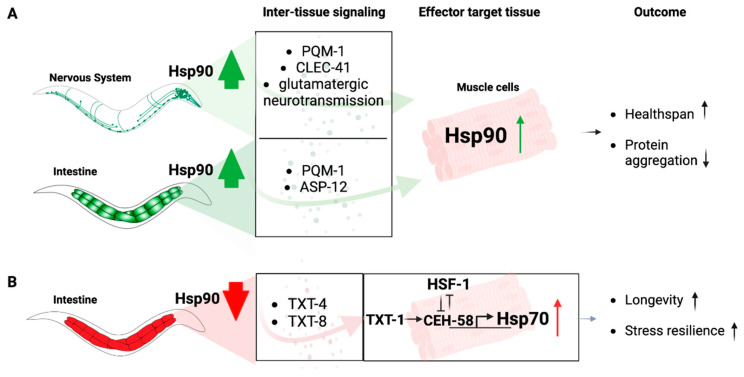
Transcellular chaperone signaling pathways. (**A**) Overexpression of Hsp90 in the nervous system mediates upregulation of Hsp90 in muscle cells via PQM-1, CLEC-41 and glutamatergic neurotransmission. Overexpression of Hsp90 in the intestine relays the signal to upregulate Hsp90 in muscle cells via PQM-1 and ASP-12. The transcription factor regulating Hsp90 in muscle cells in response to TCS has not been determined. The organismal consequences are increased health span and reduced protein aggregation in the muscle tissue. (**B**) Knockdown of Hsp90 in the intestine relays the signal to muscle cells via the secreted lipases TXT-4 and TXT-8. There, TXT-1 signals to the transcription factor CEH-58 to induce Hsp70 expression, resulting in increased longevity and stress resilience. HSF-1 functions as a suppressor of this process.

## Data Availability

Not applicable.

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
