# Peer review of "Organismal Roles of Hsp90"

_biomolecules, 2023, doi:10.3390/biom13020251_

Round 1

Reviewer 1 Report

This REVIEW ARTICLE highlight the organismal roles of Hsp90 in different multicellular model systems. While I find this study interesting, I have a few suggestions.

1. The authors should add some information about the structure of Hsp90, and it is better to use a construction diagram to make it more intuitive.

2. The authors introduced the involvement of Hsp90 in immune responses, however, I think the current content is relatively superficial, and the content of HSP90's immunity to the host needs to increase.

3. It is very interesting to clarify the role of pathogens-derived HSP90, such as malaria, Toxoplasma gondii, Schistosoma, fungal, etc.

4. In the review, the knowledge chain should be displayed in the form of a mechanism diagram and/or table, making the readers more quickly see the key information.

Author Response

We thank the reviewer for their comments. Please see the attachment for our response.

Reviewer 2 Report

This review article discusses the multiple functions of Hsp90 in multicellular organisms.  It is very a very useful compilation that covers a wide range of topics and and is worthy of publication. A few minor changes will make it more readable. 

Minor points:

There are a number of grammatical errors, such as in line 12 of the abstract, which may be missing a comma after Hsp90. 

In the introduction (line 29) as well as the abstract (line 9), the author states that the abundance of Hsp90 is needed for chaperoning signaling kinases and steroid hormone receptors.  But the remainder of the article focuses more on other function.  Perhaps statements that emphasizes that Hsp90 is needed for more than just protein kinases and steroid hormone receptors would be more appropriate.

In Figure 1 legend, it might be more to list the species names of mice and zebrafish to make it more consistent.

Line 88.  Please clarify whether deletion of Hsp90a2 has no overall effect versus no effect on muscle development.

Line 94.  Should that be HSP90AB1 mRNA?  It is not clear what organism is being discussed.

Line 101.  Should it be Wee-1 kinase functionality to be more consistent?

Line 194  Should be transcription factors…And the sentence starting with vice versa (line 196) is confusing.

Line 217, it would be helpful to just say heat shock rather than HS.

Section starting line 256. It would be helpful to specific whether this is Hsp90 alpha or beta.

Line 297 could reorder sentence to be more clear that R protein activation is Hsp90 dependent.

Author Response

I thank the reviewer for their comments and have addressed them in the attachment.

Round 2

Reviewer 1 Report

Most response are suitable. It should be noted that the HSP90 has different subunit, and some work just studies one of the subunits. It cannot be generally considered to be distinguished by HSP90. Regarding the previous round of opinions, Related To Response 4. I suggest the author needs, at least to, make a mechanism diagram about the activation and/or regulatory role of HSP90 on the signal pathway in the review.

Author Response

I have now added a new Figure (Figure 2) showing the intercellular signaling pathways induced by Transcellular Chaperone Signaling in C. elegans.